# Behҫet’s Disease, and the Role of TNF-α and TNF-α Blockers

**DOI:** 10.3390/ijms21093072

**Published:** 2020-04-27

**Authors:** Tim van der Houwen, Jan van Laar

**Affiliations:** Section of Clinical Immunology, Departments of Internal Medicine and Immunology, ErasmusMC, 3015 GD Rotterdam, The Netherlands; t.vanderhouwen@erasmusmc.nl

**Keywords:** Behçet’s disease, TNF-α, TNF-α blockers, review

## Abstract

In this both narrative and systematic review, we explore the role of TNF-α in the immunopathogenesis of Behçet’s disease (BD) and the effect of treatment with TNF-α blockers. BD is an auto-inflammatory disease, characterized by recurrent painful oral ulcerations. The pathogenesis of BD is not yet elucidated; it is assumed that TNF-α may play a key role. In the narrative review, we report an increased production of TNF-α, which may be stimulated via TLR-signaling, or triggered by increased levels of IL-1β and IFN-γ. The abundance of TNF-α is found in both serum and in sites of inflammation. This increased presence of TNF-α stimulates T-cell development toward pro-inflammatory subsets, such as Th17 and Th22 cells. Treatment directed against the surplus of TNF-α is investigated in the systematic review, performed according to the PRISMA guideline. We searched the Pubmed and Cochrane database, including comparative studies only. After including 11 studies, we report a beneficial effect of treatment with TNF-α blockers on the various manifestations of BD. In conclusion, the pivotal role of TNF-α in the immunopathogenesis of BD is reflected in both the evidence of their pro-inflammatory effects in BD and in the evidence of the positive effect of treatment on the course of disease in BD.

## 1. Introduction

Behçet’s disease (BD) is an auto-inflammatory disease, characterized by recurrent painful oral ulcerations. Diverse accompanying symptoms include uveitis, genital ulcerations, and skin manifestations such as erythema nodosum or acneiform pustular lesions. Less common complications include arthritis, gastro-intestinal, neurologic, or vascular involvement [1]. The clinic of BD is a reflection of the hyperinflammatory response to external triggers in a genetically susceptible host in which tumor necrosis factor alpha (TNF-α) may play a pivotal role. BD is proposed to be categorized more toward the autoinflammatory spectrum rather than autoimmune disorders because of the clinically overlapping symptoms with other autoinflammatory diseases, the enhanced inflammatory response, and overexpression of proinflammatory cytokines, such as TNF-α [2]. Additionally, there do not seem to exist any relevant or functional auto-antibodies in BD as in auto-immune disease. In this both narrative and systemic review, we aimed to elucidate the role of TNF-α in BD, focusing on the role of TNF-α in the pathogenesis and as a target for treatment.

## 2. Immuno-Pathogenesis of BD with Emphasis on TNF-α

The pathogenesis of BD is not yet elucidated. It is assumed that the inflammatory response in BD is caused by stimulation of the innate immune system triggered by an external danger signal. In BD, patients demonstrate a hyperactive adaptive immune system characterized by a disturbed Th1/Th2 balance, expansion of Th17 cells, and decreased regulatory T cells. Subsequently, there is a surplus of pro-inflammatory cytokines of which TNF-α may play a key role explaining the clinical symptoms [3]. This assumption is fed by a number of observations: Elevated serum and tissue TNF-α levels in BD patients and the beneficial effect of treatment with TNF-α blocking agents [4].

TNF-α is one of the 20 members of the tumor necrosis factor superfamily (TNFSF), glycoproteins with transmembrane regions that can also be expressed in secreted form. TNF-α is mainly secreted by active macrophages, but also by monocytes, endothelial cells, neutrophils, smooth muscle cells, lymphocytes, astrocytes, and adipocytes [5]. Also in BD, macrophages are not the only source of TNF-α production. Both CD4+ and CD8+ T-cells isolated from the intestinal mucosa of BD patients produced large amounts of TNF-α after stimulation with PMA [6]. Additionally, natural killer cells from BD patients produce increased levels of TNF-α in periods with active disease [7].

TNF-α production is triggered by infection and inflammation and also after trauma such as burns, myocardial infarction, or heart failure. Stimulating factors include LPS, microbiological antigens, immune complexes, enterotoxins, superantigens, C5a anaphylatoxin, IL-1, IFN-γ, GM-CSF, TGF-β, and autocrine mechanisms by TNF-α itself [8]. In BD, increased TNF-α production may be triggered via an altered TLR-response. In BD, evidence is reported for an upregulation of several TLRs. Increased expression of TLRs 2, 3, 4, and 8 has been demonstrated in PBMCs, and increased mRNA expression of TLR2 and TLR4 has been demonstrated in oral tissue from BD patients [9,10]. After TLR activation, the transcription factor NF-κB is activated, resulting in the production of pro-inflammatory cytokines like TNF-α [11]. Stimulation of the upregulated TLRs with their cognate ligand in these patients did not show an elevated TNF-α response, but this may be hampered by the immunosuppressive medication used (corticosteroids and azathioprine) [10].

TNF-α production in BD may also be stimulated by elevated levels of IFN-γ (observed in both serum and oral ulceration) or elevated levels of IL-1 (serum) [12,13,14]. 

The important role of TNF-α in the pathogenesis of BD is again emphasized by numerous studies that show elevated levels of this cytokine in the serum of BD patients [15,16,17,18,19,20,21,22]. In addition, evidence is reported for a positive correlation between TNF-α levels and disease activity [14,19,21], although other studies did not observe such an association [15,17,20]. These differences may be explained by patient characteristics (ethnicity) or medication usage (patients were on immunosuppressive medication in all studies).

TNF-α is abundant in various tissues of BD patients. For example, in oral ulcers of patients with BD, an increase in TNF-α was shown [12]. In addition, in the aqueous humor of BD patients with uveitis, elevated TNF-α levels were observed, compared to healthy controls or BD patients without uveitis [23,24]. 

In addition to this, in intestinal lesions of BD, elevated mRNA expression of TNF-α was reported [25,26,27]. 

Other studies have not demonstrated an association with TNF-α levels in affected tissues. For example, in the synovial fluid of BD patients with arthritis, lower levels of TNF-α than in rheumatoid arthritis patients are reported, possibly reflecting the less severe, non-erosive form of arthritis in BD [28,29]. In neuro-Behçet, no increase in TNF-α in cerebrospinal fluid was observed [30]. 

High levels of TNF-α causes shedding of the TNFR1 [31]. This soluble TNF-receptor 1 (sTNFR1) may act as a neutralizing agent for the pro-inflammatory effects of TNF-α [32]. This negative control is also reported in BD patients; elevated levels of sTNFR1 and sTNFR2 are observed in both the serum and synovial fluid of BD patients with active disease [14,33,34]. 

TNF-α exerts its pro-inflammatory function via binding to either tumor necrosis factor receptor 1 (TNFR-1) or 2 (TNFR-2), which are able to trigger different signal pathways. The transmembrane form of TNF-α predominantly activates TNFR-2, whereas the soluble form generally activates TNFR-1. TNFR-1 induces pro-inflammatory cytokine production via the NFκB pathway, or apoptosis via activating caspase 8 [35]. TNFR-2 signaling is less well understood, but enables cell activation, migration, and proliferation, mainly via activating the NFκB pathway [36,37].

An example of TNF-α-induced cell proliferation is displayed in the expansion of the Th17 compartment, as is reported in BD. The co-culturing of Th0 cells with various pro-inflammatory cytokines was studied, with the results directed toward IL-1β and TNF-α, to be the pivotal cytokines in BD for differentiating a T helper cell into an IL-17-producing cell [38]. 

Naïve CD4^+^ T cells are able to differentiate into IL-22 and TNF-α-producing cells, the so-called Th22 subsets, which develop along a distinct pathway from the Th1, Th2, or Th17 pathway. In the presence of both IL-6 and TNF-α, these Th22 cells arise. Due to expression of the skin-homing receptors CCR4 and CCR10, a role in inflammatory skin diseases is hypothesized [39,40,41]. In BD patients with uveitis, an increase in Th22 cells was observed in serum and aqueous humor [42]. The function of these cells was suppressed by anti-TNF-α treatment (infliximab) in vitro. Furthermore, co-culturing with anti-TNF-α and anti-IL6 antibodies (separate and combined) prevents T cells from BD patients to differentiate into Th22 cells [42]. In addition to this, increased expression of IL-22 after stimulation of PBMCs is shown in BD patients with active uveitis, and another group reported increased Th22 cells in the peripheral blood of BD patients with mucocutaneous disease [43,44].

Thus, increased levels of TNF-α stimulate T-cell development toward pro-inflammatory subsets.

### Genetics of BD

Genetic predisposition to BD is considered to be polygenetic. Genome-wide association studies (GWAS) confirmed the strong association with HLA-B51 and added IL10 and IL23R-IL12RB2 variants, which were associated with Turkish and Japanese populations [45,46]. In addition to this, susceptibility loci for BD at CCR1-CCR3, STAT4, KLRK1-KLRC4, and ERAP1 were reported [47]. ERAP-1 is responsible for trimming peptides that are loaded onto the HLA protein. Polymorphisms of both ERAP-1 and HLA could affect T- and NK-cell activation [48], which are known producers of TNF-α in BD.

Polymorphisms of the TNF-α gene associated with BD are shown in a meta-analysis of 16 studies. In Asian BD patients, polymorphisms in -308A/G and -857T/C were associated, which are both associated with increased levels of TNF-α transcription [49,50,51]. In Caucasian BD patients, the -238A/G and -1031C/T polymorphisms were more frequent; in addition, for the latter, there is evidence for an increased level of TNF-α transcription [50,51]. 

Recently, one new study was published confirming the association between Iranian BD patients and the 857T/C polymorphism [52]. Therefore, there is evidence for a genetic background for the overproduction of TNF-α in BD. 

## 3. Anti-TNF-α Treatment in BD

The ultimate confirmation of the pivotal role of TNF-α is the beneficial effect of treatment with TNF-α blocking agents on the clinical symptoms in BD. To study this effect, we performed a systematic review.

In this review, we will focus on infliximab, adalimumab, and etanercept. Infliximab (IFX) and adalimumab (ADA) are both IgG1 monoclonal antibodies, which bind to TNF-α, preventing it from activating its receptor. IFX is a mouse/human chimeric antibody, while ADA is a completely humanized antibody. Etanercept (ETC) is a soluble p75 TNF receptor fusion protein that consists of two p75 TNF receptors bound to the Fc portion of immunoglobulin G [53]. 

Given the large numbers of clinical trials and observational studies on this topic, we decided to narrow our search to comparative studies only. This was decided to increase the quality of the included studies and, therefore, the impact of our systematic review. BD diagnosis was defined according to the criteria of the International Study Group of BD [54]. Methods of this systematic review are described in detail in the Appendix A.

## 4. Systematic Review Into Anti-TNF-α Treatment

The reference flow is illustrated in Figure 1. The Pubmed search yielded 113 results, and searching the Cochrane database yielded 202 results. 

After reading the title and abstract, 11 comparative studies with TNF-α blocking agents in Behçet’s disease were selected for further reading. The search was validated by a second reviewer, adding no discrepancies.

Four studies were prospective trails, seven were retrospective trails. In the retrospective trials, five compared TNF-α blocking agents with disease-modifying antirheumatic drugs (DMARDs), and two retrospective trails compared the starting time of IFX. 

The quality assessment of the included clinical trials and observational studies is reported in Appendix A. Melikoglu et al. [55] was judged as low risk of bias, Markomichelakis et al. [56], Zou et al. [57], and Martin-Varillas et al. [58] were judged at high risk of bias. The randomization process in the latter three caused this high risk of bias. In both Markomichelakis et al. and Martin-Varillas et al., there was no randomization, and the treatment option was determined based on a shared decision. In Zou et al., no details about the randomization process are described. 

All observational studies scored a low risk of bias, with the quality assessment in Miyagawa et al. [59] scoring 7/9 and the other studies scoring 8/9. It should be noticed that the number of patients included is not taken into account in the quality assessment. 

### 4.1. Efficacy of Treatment

The results of the four prospective trials are summarized in Table 1. Overall, treatment with anti TNF-α agents shows a positive effect on a variety of disease manifestations of BD. 

Melikoglu et al. [55] is the only trial comparing TNF-α blocking treatment (i.e. ETC) with placebo. This trial only included male patients (because BD is thought to be more severe in men) and compared the treatment of 20 patients with ETC 25 mg, twice a week for four weeks, to placebo treatment. This study was designed to evaluate possible suppression of the pathergy and monosodium urate (MSU) test by ETC. Intradermal injection with MSU tests skin hyperreactivity, in analogy with the pathergy test. A suppressive effect was not detected, but a significant decrease in nodular lesions and oral ulceration in the ETC group was noticed. 

Markomichelakis et al. [56] treated 22 BD patients with uveitis (35 eyes) with either a single gift of IFX or corticosteroids (1 g iv methylprednisolone for 3 d or 4 mg triamcinolone intra-vitreal). This study also did not meet their primary objective; there was no difference in the decrease in logMAR transformed visual acuity. In one of the secondary outcomes, a significantly larger decrease in total inflammation score in the eyes in the IFX group compared to the corticosteroid group was reported. The total inflammation score consists of the counting of cells in the different segments of the eye and of the presence of vasculitis, papillitis, and retinal cystoid macular edema. 

Both Zou [57] et al. and Martin-Varillas [58] et al. compare different dosages of anti-TNF-α treatment. Both groups provide evidence for a comparable efficacy in treatment with a lower or reduced dose. 

Zou et al. compared IFX dosed at 3.5 mg/kg with IFX dosed at 5 mg/kg administered at 0, 2, and 6 w in 20 patients with intestinal BD. After 30 w of follow up, at both primary outcome (corticosteroid-free clinical remission) and secondary outcome (endoscopic mucosal healing), no significant differences were reported between both groups. Martin-Varillas et al. studied optimizing the dose of ADA in BD uveitis patients in remission on ADA treatment. Forty-two patients with 40 mg/2 w were compared to 23 patients in which, every 3 months, the dosing interval was prolonged, initially every 3 w, and then every 4, 6, and 8 w up to discontinuation. They report a comparable number of relapses of uveitis per 100 patients per year, but a significant decrease in costs. 

There are three retrospective trials comparing IFX with a DMARD in BD patients with uveitis (Table 2).

Tabbara et al. [60] and Yamada et al. [61] both compared DMARD treatment (Tabbara et al., corticosteroids, cyclosporin A, methotrexate, and azathiopirin; Yamada et al., cyclosporine A) with IFX 5 mg/kg treatment. Tabbara et al. differed in a longer follow up (30–36 months) from Yamada et al. (6 months). Both these studies reported a significant decrease in the number of relapses of uveitis in patients treated with IFX compared to DMARDs. To compare the number of relapses, we have calculated the number of relapses per 6 months, which is, in Tabbara et al., 1.6 for the DMARD group and 0.3 for the IFX group. These numbers are comparable to Yamada et al., 1.2 in the DMARD group and 0.4 in the IFX group. Both studies reported the best corrected visual acuity (BCVA) as a secondary outcome. Tabbara et al. observed a significantly larger percentage of patients with a good BCVA (defined as 20/50 or better) in the IFX group. Yamada et al. observed a comparable percentage of improved and unchanged BCVA between the IFX group and the DMARD group. 

Takeuchi et al. [62] compared IFX monotherapy with IFX combined to colchicine in 14 patients with BD uveitis. After a median follow-up period of 25 months, no significant difference in uveitis relapse per 6 months was noticed. In addition, measured in BCVA, no added value of colchicine treatment was found. 

In 70 BD patients with vascular involvement, Emmi et al. [63] compared ADA with DMARD treatment. Measured by ultrasound, the percentage of complete or partial response of the vascular lesions was significantly higher in the ADA treatment group. In addition to this, the mean time to achieve the vascular response was almost 3 w shorter in the ADA treatment group. 

Miyagawa et al. [59] compared the effectiveness of anti-TNF-α treatment with corticosteroids. Next to this, the additive value of corticosteroids to anti-TNF-α treatment was studied. In total, 71 patients with intestinal BD were followed for 12 months. Efficacy was studied by endoscopic evaluation of intestinal ulceration. The percentage of cure rate was significantly better with TNF treatment compared to corticosteroids (*p* = 0.0029). Adding corticosteroids to TNF treatment caused no significant further improvement in cure rate of intestinal ulceration. This is also observed for the percentage of improved intestinal ulcers. 

We have reviewed two retrospective studies comparing the starting time of IFX treatment in patients with BD uveitis (Table 3). Keino et al. [64] compared, in 13 BD patients with uveitis, IFX treatment starting within 18 months of the onset of ocular disease to IFX treatment starting after 18 months. In a number of relapses, no significant differences were observed between early or late IFX treatment. Early treatment did show a significant decrease in disc and retinal leakage scores; in addition to this, BCVA was significantly better in the early treatment group. This last result should be interpreted with caution, because baseline characteristics of the early treatment group regarding BCVA were also significantly better compared to the late treatment group. 

Guzelant et al. [65] compared treatment with IFX for uveitis with their own historical cohort (before and after January 2013). In the new cohort, they initiated IFX treatment significantly earlier in the disease course (36 vs. 72 months). This resulted in significantly better results in BCVA (stable visual acuity in 64% in new cohort vs. 28% of patients in old cohort) and in a significant decrease in relapses (7% in new cohort vs. 53% in old cohort). 

### 4.2. Adverse Events

TNF-α blocking treatment is mostly associated with infections, allergic reactions, or skin reactions and malignancies. 

In this systematic review, in all articles, 313 patients were treated with one of three different anti-TNF-α agents. Seventeen patients were described with infections (5%), majority viral (nine patients), six patients with bacterial infections, and two cases of pulmonary tuberculosis (despite screening and INH treatment). Eight patients suffered allergic reactions, all after infusion of IFX (2.5%). One malignancy (lymphoma) was described, and 12 patients experienced skin reactions after treatment with an anti-TNF-α agent. No deaths due to adverse events were observed.

## 5. Discussion

In this both narrative and systemic review, we discuss the role of TNF-α in Behçet’s disease. TNF-α has a pivotal role in the immunopathogenesis of BD. Here, we report of increased production of TNF-α by macrophages, CD4+ and CD8+ T-cells, and NK-cells in patients with BD [6,7]. This increased production may be stimulated via TLR-signaling, or triggered by the increased levels of IL-1β and IFN-γ, which are observed in serum of BD patients [9,10,12,13,14]. The abundance of TNF-α is found in both serum and in sites of inflammation such as oral ulceration, intestinal ulceration, and the aqueous humor of patients with BD uveitis [14,15,16,17,18,19,20,21,22,23,24]. In serum, there may be a positive correlation between the levels of TNF-α and disease activity, although there is also evidence that argues against this. The inconsistency may be explained by the heterogeneity in patient cohorts and immunosuppressive medication used. The increased presence of TNF-α in serum and tissue stimulates T-cell development toward pro-inflammatory subsets, such as Th17 and Th22 cells [38,42].

In the systematic review, we add data about the effects of treatment with TNF-α blocking agents on the course of BD. We observe an overall beneficial effect of treatment with TNF-α blocking agents on the various manifestations of BD. Unfortunately, this observation is not based on high-quality prospective randomized control trials. Only four prospective trials could be included, where three were assessed at high risk of bias. This was caused by demerits in the randomization process. Despite the risk of bias, all studies display the value of treatment with anti-TNF-α agents in BD. 

The retrospective observational studies included in this systematic review were all of high quality. Four studies compared anti-TNF-α treatment with DMARDs. These studies observed significant beneficial effects on uveitis (decrease in mean number of relapses), intestinal BD (increased ulcer cure rate), and vascular BD (increase in vascular response on ultrasound examination). Takeuchi et al. showed no additional value of colchicine treatment next to IFX in a small cohort of BD uveitis patients. 

Early initiating of IFX treatment was evidenced to be effective in increasing BCVA and in decreasing the number of relapses by Guzelant et al. In a small cohort of 13 BD patients, Keino et al. observed no difference in the number of relapses of uveitis in patients starting IFX treatment within 18 months of onset of ocular disease or IFX treatment starting after 18 months.

Despite the risk of bias in the prospective trials, we can conclude that treatment with anti-TNF-α agents (especially IFX and ADA) has a beneficial effect on the course of disease in BD. This is evidenced in uveitis, intestinal, and vascular BD. The clinical effects were achieved with a relatively low incidence of adverse effects. Infections (5%) are most common, followed by allergic reactions (2.5%). In 313 patients, one malignancy was reported (lymphoma). 

The broad spectrum of symptoms that are alleviated by treatment with anti-TNF-α agents confirms the major role of TNF-α in the diverse manifestations of BD. Blocking of TNF-α downregulates pro-inflammatory T-cell subsets and restores the memory B-cell compartment [66,67,68]. Next to the immunologic effects of anti-TNF-α treatment, the beneficial effect on vascular BD disease suggests an effect on the vascular endothelial cells, but this is not yet evidenced in BD. 

There are several limitations to our systematic review. Due to the heterogeneity of outcome measurements, we were unable to perform a meta-analysis. In addition, prefiltering on comparative studies caused a relatively low number of studies to be eligible for inclusion. Next to this, the included retrospective studies have the potential risk of selection bias. 

Nevertheless, our systematic review has several strengths. First, we only included comparative studies to increase the quality of our review. Second, all retrospective studies included were of high quality. Third, this is the first review study specifically aimed at TNF-α in BD, describing the role in pathogenesis and the clinical effects of treatment with TNF-α blocking treatment. 

In conclusion, we present a both narrative and systematic review focused on TNF-α in BD. The pivotal role of TNF-α in the immunopathogenesis of BD is reflected by the increased production of TNF-α by macrophages, CD4+ and CD8+ T-cells, and NK-cells and by the increased levels of TNF-α in serum and in inflamed tissue. Treatment directed against this cytokine is evidenced to positively affect the course of disease in BD. 

## Figures and Tables

**Figure 1 ijms-21-03072-f001:**
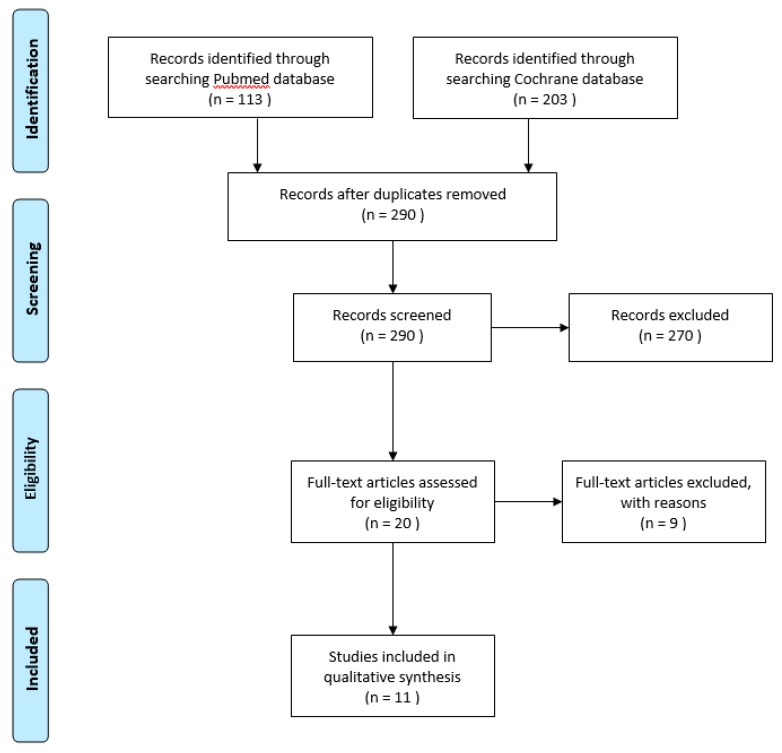
Systematic review flow diagram according to PRISMA guidelines.

**Table 1 ijms-21-03072-t001:** Prospective comparative trials with anti TNF-α agents.

Author (Trial)	Year	Patients	Study	Duration of Follow Up	Numbers Included	Outcome	Treatment	Comparator	*p*-Value	Adverse Events
Melikoglu	2005	Muco- cutaneous BD	Treatment	3 months	Treatment group	Primary outcome	decrease in pathergy positivity			Treatment
			ETC (25 mg, twice a week)		20 patients	suppression of pathergy and MSU	58%	58%	ns	1 diarrhoe
							decrease in MSU, mm2			
							1144	944	ns	
			Comparator		Comparator group	Secundary outcome	% patients free of nodular lesions			Comparator
			Placebo		20 patients	difference in mean numbers of mucocutaneous lesions and swollen joints	85%	25%	*p* = 0.0002	1 elevated liver enzymes
							% patients free of oral ulceration			
							45%	5%	*p* = 0.0017	
Markomichelakis	2011	BD uveitis	Treatment	1 month	Treatment group	Primary outcome	decrease in logMAR transformed VA			Treatment
			IFX 5 mg/kg single gift		19 eyes	visual acuity, logMAR transformed	1.2 -> 0.5	1.6 -> 0.7	ns	None
			Comparator		Comparator group	Secundary outcome	decrease in total inflammation score			Comparator
			CCS*		8/8 eyes	ao total inflammation score	largest decrease in IFX on day 14		*p* = 0.010	ocular hypertension in 4 triamcinolon treated eyes
Zou	2017	Intestinal BD	Treatment	30 weeks	Treatment group	Primary outcome	corticosteroid-free clinical remission, no (%)			Treatment
			IFX 3.5 mg/kg		10 patients	corticosteroid-free clinical remission	4 (40%)	6 (60%)	*p* = 0.371	1 eczema; topical therapy and 1 common cold
			Comparator		Comparator group	Secundary outcome	mucosal healing at week 14, no (%)			Comparator
			IFX 5 mg/kg		10 patients	endoscopic mucosal healing	6 (60%)	6 (60%)	*p* = 1.0	none
Martin Varillas	2018	BD uveitis	Treatment	34 months	Treatment group	Primary outcome	relapses, n (per 100 patients/year)			Treatment
			ADA optimized dose		23 patients	relapse of uveitis	2 (3.0)	4 (4.4)	*p* = 0.66	None
			Comparator	26 months	Comparator group	Secundary outcome	costs (mean), euros per year			Comparator
			Non optimized ADA		42 patients	costs	6101,25	12339,48	*p* < 0.01	lymphoma, pneumonia, Ecoli, local reaction (1 each)

Abbreviations used: ADA adalimumab, CCS corticosteroids, ETC etanercept, IFX infliximab, MSU monosodium urate, VA visual acuity; * 3 d iv 1 g methylprednisolone or 4 mg triamcinolone intra-vitreal.

**Table 2 ijms-21-03072-t002:** Retrospective comparative trials with anti TNF-α agents and DMARDs.

Author (Trial)	Year	Patients	Study	Duration of Follow Up	Numbers Included	Outcome	Treatment	Comparator		*p*-Value	Adverse Events
Tabbara	2008	BD uveitis	Treatment	36 months	Treatment	Primary outcome	Mean no. of relapses (range)				Treatment
			IFX*, CCS, AZA		10 patients	number of relapse	1.2 (0 to 4)	6.3 (4 to 7)		*p* < 0.001	2 mild infusion reactions, 1 infection (perianal abces)
			Comparator	30 months	Comparator	Secundary outcome	Patients with good BCVA^¥^ (%)				Comparator
			CCS, CsA, AZA, MTX		33 patients	BCVA	50%	6%		*p* = 0.0059	4 hypertension, 5 renal function decrease, 3 elevated liver enzymes, 7 hyperglycemia
Yamada	2010	BD uveitis	Treatment	6 months	Treatment	Primary outcome	Mean no. of relapses (SD)				Treatment
			IFX**		17 patients	number of relapse	0.4 (±1.0)	1.2 (±1.2)		*p* < 0.05	9 skin symptoms (eruptions, itching, atopic dermatitis); topical therapy
			Comparator		Comparator	Secundary outcome	Improved/unchanged BCVA (%)				Comparator
			CsA		20 patients	BCVA	97%	93%		ns	1 neurologic symptoms and renal toxicity; dose reduction
Takeuchi	2012	BD uveitis	Treatment	25 months	Treatment	Primary outcome	Mean no. of relapses/6 months (SD)				Treatment
			IFX**		7 patients	number of relapse	0.22 (±0.28)	0.18 (±0.19)		ns	1 infusion reaction
			Comparator	33 months	Comparator	Secundary outcome	Improved/unchanged BCVA (%)				Comparator
			IFX**, colchicine		7 patients	BCVA	100%	83.3%		ns	CMV infection; cured by valganciclovir
Emmi	2018	Vascular BD	Treatment	26 months	Treatment	Primary outcome	Complete or partial response, no (%)				Treatment
			ADA		35 patients	vascular response (ultrasound)	34 (97.1%)	23 (66%)		*p* = 0.001	2 urticarial skin rash, 1 pneumonia, 1 HSV reactivation
			Comparator		Comparator	Secundary outcome	Mean time (SD), weeks				Comparator
			DMARDs		35 patients	time to achieve vascular response	3.7 (±1.7)	6.3 (±1.2)		*p* < 0.0001	2 adverse events
Miyagawa	2019	Intestinal BD	Treatment	12 months	Treatment	Primary outcome	Patients with cured ulcer, no (%)				Treatment
			TNF-i with CCS		20 patients	ulcer cure rate (endoscopic)	12 (60%)	13 (45%)	3 (13.6%)		1 bacterial infection
			Comparator		Comparator	Secundary outcome	Patients with improvement of ulcer, no (%)				Comparator
			TNF-i without CCS		29 patients	ulcer improvement rate	12 (60%)	15 (57%)	6 (27%)		none described
			Comparator		Comparator						Comparator
			CCS		22 patients						

Abbreviations used: ADA adalimumab, AZA azathioprine, BCVA best corrected visual acuity, CCS corticosteroids, CsA cyclosporin A, CMV cytomegalovirus, DMARDs disease-modifying anti-rheumatic drugs, HSV herpes simplex virus, IFX infliximab, MTX methotrexate, SD standard deviation. * = (5 mg/kg) day 1 and at 2, 4, 6, 8, and 10 weeks; ** = (5 mg/kg) day 1 and at 2, 4, 6, and 8 weeks; ¥ = 20/50 or better.

**Table 3 ijms-21-03072-t003:** Retrospective trials comparing infliximab in starting time.

Author (Trial)	Year	Patients	Study	Duration of Follow Up	Numbers Included	Outcome	Treatment	Comparator		*p*-Value	Adverse Events
Keino	2017	BD uveitis	Treatment	24 months	Treatment	Primary outcome	Mean no. of relapses/6 months (SD)				Treatment
			Uveitis <18 months		6 patients	number of relapse	no significant differences				5 viral infection
			Comparator		Comparator	Secundary outcome	Vascular leakage score				Comparator
			Uveitis >18 months		7 patients	ao vascular leakage score	significant decrease in <18mcompared to >18m after year 1 and 2				1 viral infection, 1 bacterial pharyngitis responding to antibiotic therapy
Guzelant	2017	BD uveitis	Treatment	12 months	Treatment	Primary outcome	Stable VA after IFX in R eye, n (%)				Treatment
			After January 2013		14 patients	BCVA	9 (64)	12 (28)		*p* = 0.01	3 allergic reactions, 1 pulmonary TBC
			Comparator	40 months	Comparator	Secundary outcome	Number of relapses, n (%)				Comparator
			Before January 2013		43 patients	number of relapse	1 (7)	23 (53.4)		*p* = 0.002	2 allergic reactions, 1 pulmonary TBC, 1 lung nodule

Abbreviations used: BCVA best corrected visual acuity, IFX infliximab, TBC tuberculosis, SD standard deviation.

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
