# Peer review of "Behҫet’s Disease, and the Role of TNF-α and TNF-α Blockers"

_ijms, 2020, doi:10.3390/ijms21093072_

Round 1

Reviewer 1 Report

This is a timely and comprehensive review of the role of TNF and its inhibition in patients with Behcet's Disease (BD). As these drugs are used extensively in BD such an analysis is relevant. However, there are some points that should be addressed;

(1) lines 92-100 discuss the receptors for TNF and this should be extended, in particular with greater discussion on soluble forms of these receptors which can inhibit TNF activity particularly sTNF. Moreover, as STNFR are induced by TNF activity itself they act as a dominant negative control of the system. This should be made clearer.

(2) a section on the different forms of TNF inhibitors should be added. Etanercept is a fusion protein of TNFR and Ig constant region, infliximab, a mouse:human chimeric antibody and adalimumab, a fully human antibody. The potential different modes of action ie etanercept only blocks sTNF may be relevant to the findings.

(3) In the discussion, the authors should be more definitive on how they believe TNF inhibition is working and is it more effective with some manifestations of BD than other. If the major effect is to prevent activation of vascular endothelial cells, any effect on T cell populations made be secondary. 

(4) line 79 - seems misplaced and not relevant and could be removed.

Author Response

Dear Reviewer,

Thank you for the opportunity to revise our manuscript. We were happy to receive these suggestions to improve our manuscript. All changes are highlighted in yellow. Below you will find our point-by-point responses.

With kind regards,

Tim van der Houwen, MD

(on behalf of all authors)

Reviewer 1
This is a timely and comprehensive review of the role of TNF and its inhibition in patients with Behcet's Disease (BD). As these drugs are used extensively in BD such an analysis is relevant. However, there are some points that should be addressed;

(1) lines 92-100 discuss the receptors for TNF and this should be extended, in particular with greater discussion on soluble forms of these receptors which can inhibit TNF activity particularly sTNF. Moreover, as STNFR are induced by TNF activity itself they act as a dominant negative control of the system. This should be made clearer.

This is an interesting point. We thank the reviewer for noticing and have added the following (line 82-86):

High levels of TNF-a causes shedding of the TNFR1.39 This soluble TNF-receptor 1 (sTNFR1) may act as a neutralizing agent for the pro-inflammatory effects of TNF-a.40 This negative control is also reported in BD patients; elevated levels of sTNFR1 and sTNFR2 are observed in both serum and synovial fluid of BD patients with active disease.13 41,42

(2) a section on the different forms of TNF inhibitors should be added. Etanercept is a fusion protein of TNFR and Ig constant region, infliximab, a mouse:human chimeric antibody and adalimumab, a fully human antibody. The potential different modes of action ie etanercept only blocks sTNF may be relevant to the findings.

We agree with the reviewer and have added a small section (129-134).

In this review we will focus on infliximab, adalimumab and etanercept. Infliximab (IFX) and adalimumab (ADA) are both IgG1 monoclonal antibodies, which bind to TNF-α, preventing it from activating its receptor. IFX is a mouse/human chimeric antibody, while ADA is a completely humanized antibody.
Etanercept (ETC) is a soluble p75 TNF receptor fusion protein that consists of two p75 TNF receptors bound to the Fc portion of immunoglobulin G.

(3) In the discussion, the authors should be more definitive on how they believe TNF inhibition is working and is it more effective with some manifestations of BD than other. If the major effect is to prevent activation of vascular endothelial cells, any effect on T cell populations made be secondary.

We have expanded our discussion section (line 269-273).

The broad spectrum of symptoms that are alleviated by treatment with anti TNF-α agents confirms the major role of TNF-α in the diverse manifestations of BD. Blocking of TNF-α downregulates pro-inflammatory T-cell subsets and restores the memory B-cell compartment.66,67 Next to the immunologic effects of anti TNF-α treatment, the beneficial effect on vascular BD disease suggests an effect on the vascular endothelial cells, but this is not yet evidenced in BD.

(4) line 79 - seems misplaced and not relevant and could be removed.

We agree with the reviewer and have removed this sentence.

Reviewer 2 Report

The Review “Behcet’s disease, the role of TNF-α and TNF-α blockers”, focuses on the role of TNF- α on this disease and both explore and discuss all the studies performed to elucidate these aspects.

It is well written and organized and in my opinion can be accepted in the present form.

Congratulations to the authors.

The Review “Behcet’s disease, the role of TNF-α and TNF-α blockers”, focuses on the role of TNF- α on this disease and both explore and discuss all the studies performed to elucidate these aspects.

It is well written and organized and in my opinion can be accepted in the present form.

Congratulations to the authors.

Author Response

Dear Reviewer,

Thank you for the opportunity to revise our manuscript. We were happy to receive these suggestions to improve our manuscript. All changes are highlighted in yellow. Below you will find our point-by-point responses.

With kind regards,

Tim van der Houwen, MD

(on behalf of all authors)

Reviewer 2
The Review “Behcet’s disease, the role of TNF-α and TNF-α blockers”, focuses on the role of TNF- α on this disease and both explore and discuss all the studies performed to elucidate these aspects.

It is well written and organized and in my opinion can be accepted in the present form.

Congratulations to the authors.

We would like to thank the reviewer for his comment.

Reviewer 3 Report

Suggestions for the Introduction.

Line 29:  A short discussion of autoinflammatory vs autoimmune diseases could be beneficial.

Suggestions for section 2.  Immuno-pathogenesis...

Line 72:  tumor necrosis factor alpha should be abbreviated.

Line 100:  NFKB change to NFκB.

Line 109:  Reference

This section would flow better for the reader if it were reorganized to discuss the immunology concepts together and the genetics separately or in another sub-section.

The organization of studies reviewed is okay, but grouping studies of patients with similar clinical signs (while still pointing out which are prospective and which are retrospective) might help reveal trends a little better.

The relative paucity of number of studies reviewed should be mentioned in the discussion.

Was the number of patients in each study accounted for when determining the strength of each study?

Tables should be revised so that the reader is not guessing what is meant.  Please add detailed table legends.  Be sure to indicate what 1 - a) and 1 - b) means.

Author Response

Dear Reviewer,

Thank you for the opportunity to revise our manuscript. We were happy to receive these suggestions to improve our manuscript. All changes are highlighted in yellow. Below you will find our point-by-point responses.

With kind regards,

Tim van der Houwen, MD

(on behalf of all authors)

Reviewer 3

Line 29:  A short discussion of autoinflammatory vs autoimmune diseases could be beneficial.

We agree this could be beneficial and have added an extra sentence to our introduction (line 35-40):

BD is proposed to be categorized as auto-inflammatory because of the clinically overlapping symptoms with autoinflammatory diseases, the enhanced inflammatory response and overexpression of proinflammatory cytokines, such as TNF-α.2Additionally, there don’t seem to exist any relevant or functional auto-antibodies in BD as in auto-immune disease.

Suggestions for section 2.  Immuno-pathogenesis...

Line 72:  tumor necrosis factor alpha should be abbreviated.

Line 100:  NFKB change to NFκB.

Line 109:  Reference

We would like to thank the reviewer for noticing these small mistakes. We have corrected them (line 117, 90, 92, and 101).

This section would flow better for the reader if it were reorganized to discuss the immunology concepts together and the genetics separately or in another sub-section.

We agree with the reviewer and we have moved the genetics section towards the end of the immunopathogenesis (line 109-124)

The organization of studies reviewed is okay, but grouping studies of patients with similar clinical signs (while still pointing out which are prospective and which are retrospective) might help reveal trends a little better.

In our opinion, the organization of studies as it is at the moment is optimal.
We believe it is important to clearly separate the prospective and retrospective studies to prevent confusion. Therefore we have opted to start with table 1 with the four prospective studies.
We do agree grouping of studies helps to reveal trends; therefore we have grouped the studies of IFX comparing starting time together in table 3. Also in table 2 the studies into IFX in BD uveitis are presented as first to help the reader compare these studies.

The relative paucity of number of studies reviewed should be mentioned in the discussion.

We have added this sentence to the discussion section (line 275-276):

Also, prefiltering on comparative studies caused a relatively low number of studies to be eligible for inclusion.

Was the number of patients in each study accounted for when determining the strength of each study?

The Newcastle-Ottawa scale does not take number of patients into account. We have added this sentence (line 156-157):

It should be noticed that number of patients included is not taken into account in the quality assessment.

Tables should be revised so that the reader is not guessing what is meant.  Please add detailed table legends.  Be sure to indicate what 1 - a) and 1 - b) means.

Thank you for noticing, indeed we forgot to add the table legends. We now included them into the manuscript.